# Reported adverse events following COVID-19 vaccination in gynecologic cancer patients in Thailand: A descriptive study

Sasivimon Ratree[1], Nicha Assavapokee[1], Somsook Santibenchakul[1], Nakarin Sirisabya[1], Waranyoo Phoolcharoen[2,3], Natacha Phoolcharoen[1]*

1 Department of Obstetrics and Gynecology, Faculty of Medicine, Chulalongkorn University, King Chulalongkorn Memorial Hospital, Bangkok, Thailand, 2 Center of Excellence in Plant-produced Pharmaceuticals, Chulalongkorn University, Bangkok, Thailand, 3 Department of Pharmacognosy and Pharmaceutical Botany, Faculty of Pharmaceutical Sciences, Chulalongkorn University, Bangkok, Thailand.

* phnatacha@gmail.com, Natacha.p@chula.ac.th

## Abstract

### Objective

Data on the safety of coronavirus disease 19 (COVID-19) vaccines in gynecologic cancer patients are scarce. The type of vaccine used in Thailand differs from what has been studied in other countries. This study evaluated the prevalence and characteristics of reported adverse events following COVID-19 vaccines in patients with gynecologic cancer patients.

### Method

A retrospective, single-center descriptive study was performed in patients with gynecologic cancer who received at least one dose of the COVID-19 vaccine at King Chulalongkorn Memorial Hospital, Thailand, from January 2020 to August 2021. Adverse events were collected through structured telephone interviews using a standardized questionnaire. Descriptive statistics summarized patient characteristics and adverse events. Associations with any-grade adverse events were assessed using logistic regression and Fisher's exact test for categorical variables.

### Results

Of the 294 patients interviewed, 72.8% were in remission, under surveillance, or in palliative treatment at the time of vaccination, and 17.7% were undergoing treatment. The most common adverse effects were grade 1–2 injection site reactions. One patient developed grade 3 fever and seizures 10 days after the first dose of the AstraZeneca vaccine. Between the second and fourth doses of the vaccination, the most common adverse events were grade 1–2 injection site reactions. No severe allergic reactions or grade 4 adverse events were reported. The study concluded

**Data availability statement:** The data are not publicly available due to ethical and legal restrictions, as the dataset contains potentially identifying and sensitive information from human research participants. Public sharing of the data is not permitted under the terms of approval by the Institutional Review Board of the Faculty of Medicine, Chulalongkorn University. De-identified data may be made available upon reasonable request to researchers who meet the criteria for access to confidential data. Requests should be submitted to the Institutional Review Board, Faculty of Medicine, Chulalongkorn University, via medch-ulairb@chula.ac.th.

**Funding:** The author(s) received no specific funding for this work.

**Competing interests:** The authors have declared that no competing interests exist.

that patients under 60 years of age had more adverse events than older patients (adjusted odds ratio 1.99, 95% confidence interval 1.08–3.71 p = 0.029). The treatment status did not affect adverse events. Of 283 patients who received two doses, 27.6% were infected with COVID-19.

## Conclusion

COVID-19 vaccination was generally well tolerated among gynecological cancer patients who received active anticancer therapy and those under surveillance. Younger patients frequently reported more adverse effects than older patients.

## Introduction

Coronavirus disease 19 (COVID-19) caused by severe acute respiratory syndrome coronavirus 2 (SARS-CoV-2) has emerged as a pandemic that has led to a global public health crisis. COVID-19 infection was first reported in December 2019, and the pandemic ultimately caused 6.8 million deaths worldwide [1]. Older adults and people who are immunocompromised, including patients with cancer, most often experience severe symptoms when infected with COVID-19.

Immunocompromised patients with cancer are at a higher risk of infection and severe complications, including intubation or mechanical ventilation, intensive care unit admission, and death. For example, patients with cancer had a 39% probability of being in a challenging health condition and requiring treatment in the intensive care unit compared with 8% of people without cancer (P = 0.0003) and a higher mortality rate than patients without cancer (20% versus 11%, P = 0.006) [2,3]. Two concerns are associated with the impact of ongoing cancer treatment on vaccination: treatment may cause derivative diseases and may weaken the immune response to the vaccine.

The COVID-19 vaccine was developed and studied to prevent infection by COVID-19 by stimulating the immune response to SAR-CoV-2. However, patients with cancer were often excluded from phase 3 trials of vaccine studies because they receive immunosuppressive drugs or are treated with chemotherapy [4–6].

In Thailand, the government initially introduced the Sinovac and AstraZeneca vaccines, which were acquired in February 2021 and June 2021, respectively. The government's free-of-charge COVID-19 vaccination program was provided in the first half of 2021 to healthcare workers and patients with one of seven underlying diseases. The program was subsequently offered to the general public free of charge in the late third quarter of 2021. Thailand further received a donation of one million doses of the Pfizer mRNA-based vaccine, which arrived in August 2021 from the United States. However, given limited quantities, the mRNA-based vaccines were not broadly offered until December 2021.

Although the pandemic appeared to have ended in 2022 with the Omicron variant, COVID-19 is expected to persist as a global recurring disease. To date, data are limited on the safety and tolerability of COVID-19 vaccines that can be used in patients

with gynecologic cancer who receive active anticancer therapy or are under surveillance. Moreover, the type of vaccine used in Thailand differs from what has been studied in other countries. Although vaccine hesitancy and the refusal ratio are substantially low (estimated at 10%) [7]; data on vaccine acceptance in Thai patients with cancer are limited. More data on vaccine safety in patients with cancer could increase patient confidence in receiving vaccines that protect against COVID-19.

Gynecologic cancer patients, while being a subset of immunocompromised individuals, have distinct characteristics including gender specificity, treatment regimens, and regional health behaviors that could influence vaccine response. Additionally, in our setting, gynecologic oncology patients were among the prioritized groups for early COVID-19 vaccination. This subgroup analysis adds specific insights to guide clinical counseling in similar populations. Addressing this gap may provide more tailored guidance for vaccination in this unique oncology population. This study evaluated the prevalence and characteristics of reported adverse events following COVID-19 vaccination. In addition, it aims to support patient counseling by providing evidence to reassure cancer patients regarding vaccine safety and to encourage vaccination among those who have not yet been vaccinated.

## Materials and methods

### Study design and population

After obtaining approval from the Research Ethics Committee, we conducted a descriptive, single-center study by reviewing medical records of patients with gynecologic cancer who had received at least one dose of the COVID-19 vaccine between January 2020 and August 2021. Data access for research purposes was granted in March 2022.

### Patient selection

The study excluded women younger than 18 years of age, patients with missing data, and patients who refused to participate. Demographic data and information on oncological treatment were collected from electronic medical records. 'Current treatment' was used to describe treatment received within 28 days of receiving the vaccine.

### Data collection

Adverse events following vaccination were collected via telephone interviews using standardized questionnaire administered by trained oncology staff. Patients were asked about predefined solicited adverse events within 7 days after vaccination, as well as unsolicited events up to 30 days. For some patients, adverse event data were cross-referenced with entries in the national "Mohprompt" application. Adverse events were graded in accordance with the Common Terminology Criteria for Adverse Events (CTCAE) v 5.0. For fever, if a temperature was recorded, grading followed CTCAE cutoffs (<39°C for Grade 1, 39–40°C for Grade 2, >40°C for Grade 3). If no temperature was recorded, grading was based on patient-reported severity: "felt warm but did not limit activities"=Grade 1, "interfered with daily activities or required antipyretics"=Grade 2, and "required hospital visit or medical attention"=Grade 3. Other systemic symptoms (fatigue, myalgia, headache, etc.) were graded according to CTCAE v5.0 definitions, based on their impact on daily functioning. COVID-19 infection data included patients with Antigen Test Kit who were PCR-positive at least 14 days after the second injection.

### Outcomes and statistical analysis

The primary endpoint was the occurrence of solicited adverse events within 7 days of vaccination and unsolicited events from days 8–30 post-vaccination. The secondary endpoint was the incidence of COVID-19 infection diagnosed at least 14 days after the second vaccine dose.

Statistical analysis was performed using Statistical Package for the Social Sciences software (IBM SPSS Statistics version 29). We used descriptive statistics to describe demographic data, oncologic characteristics, and vaccine safety

outcomes. Although both Fisher's exact test and the Chi-square test were initially considered, all reported p-values are based on Fisher's exact test to ensure consistency and accuracy, given the small sample sizes in several subgroups. This approach was applied when assessing the association between cancer treatment status (active treatment vs. surveillance/ palliative care) and adverse events. Binary logistic regression analysis was performed to explore the association between clinical factors and adverse event occurrence. As a sensitivity analysis, we constructed a fully adjusted logistic regression model to examine the association between patient characteristics and the occurrence of any-grade adverse events after the first dose of COVID-19 vaccine. P-values less than 0.05 were considered statistically significant.

## Results

From January 2020 to August 2021, 846 patients with gynecological cancer presented at the King Chulalongkorn Memorial Hospital. Of these patients, 294 patients were eligible for analysis. The rest were excluded mainly due to being unreachable, having received vaccination elsewhere with unavailable records, or being under 18 years of age. The baseline characteristics of the study population are provided in Table 1. The median patient age was 61.50 (21–88) years. The mean body mass index (BMI) was 24.50 (14.96–49.54). Most patients (79.6%) had an Eastern Cooperative Oncology Group (ECOG) performance status score of 0 or 1, and 51.4% of patients had at least one comorbidity.

The most frequent oncological malignancies were ovarian, fallopian tube, and primary peritoneal cancers (39.1%) followed by uterine cancer (34.7%), cervical cancer (20.7%), multiple sites of primary cancer (2.7%), vaginal cancer (2%), and vulvar cancer (0.7%). Most patients (214, 72.8%) were in remission, under surveillance, or in palliative treatment at the time of vaccination; the remaining patients (52, 17.7%) were in ongoing treatment.

Among the 294 patients who received a first dose, 283 (96.3%) patients received two doses of the vaccine, 190 (64.6%) patients received three doses, and 80 (27.2%) patients received four doses. The types of COVID-19 vaccination for the first dose were Oxford/AstraZeneca (273, 92.9%), Sinovac (12, 4.1%), Moderna (5, 1.7%), and Pfizer/BioNTech (4, 1.4%) (Table 2).

Among patients who received multiple vaccine doses, the median interval between the first and second doses was 11.6 weeks (SD 3.3). The interval between the second and third doses was 20.5 weeks (SD 11.7), and between the third and fourth doses was 20.3 weeks (SD 6.8).

The median duration of follow-up for vaccine-related adverse events was 493 (range:15–561) days. After the first dose, 80.2% of patients developed any-grade adverse events. Within 7 days post-vaccination, the most common symptoms were injection site reactions (64.3%), fever (44.2%), myalgia (14.3%), fatigue (7.8%), and headaches (4%). No grade 3–4 events or anaphylaxis were observed. The most common symptom at 7 days post-vaccination was injection site reactions (0.6%). Only one patient developed grade 3 fever and seizures at 10 days post-vaccination after receiving the AstraZeneca vaccine. This patient was diagnosed with encephalopathy attributed to stress or fever in the context of underlying pathologic brain changes from an old ischemic stroke. Vaccine-Induced Immune Thrombotic Thrombocytopenia (VITT) was systematically excluded through normal platelet count ($>150 \times 10^9$/L), absence of thrombosis on imaging (brain CT and venous Doppler), normal D-dimer, and negative anti-platelet factor 4, antibody enzyme-linked immunosorbent assay (ELISA), confirming non-thrombotic etiology per international diagnostic criteria. The patient fully recovered after supportive care (antipyretics, anticonvulsants, and neurology consultation) with no neurologic sequelae at 6-month follow-up (Table 3).

Following the second dose, the patients developed the following grade 1–2 symptoms: injection site reactions (63.3%), fever (27.2%), fatigue (6.7%), and myalgia (6.4%); only 0.4% of patients experienced injection site reactions for more than 7 days. Following the third dose, grade 1–2 symptoms included injection site reactions (68.9%), fever (30.5%), fatigue (8.4%), and myalgia (6.8%); only one patient developed axillary lymphadenopathy for more than 7 days. Following the fourth dose, grade 1–2 symptoms were injection site reactions (70%), fever (26.25%), fatigue (13.75%), and myalgia (10%), whereas 1.25% of patients developed injection site reactions, myalgia, and lymphadenopathy that lasted more

**Table 1. Baseline Characteristics (N = 294).**

| Characteristic | N (%) |
|---|---|
| **Age** | |
| 18 to <60 years | 128 (43.5) |
| ≥60 years | 166 (56.5) |
| **Median age (IQR), years** | 61.50 (52-61.5) |
| **Body mass index (kg/m²)[a]** | |
| Underweight (< 18.5) | 23 (7.8) |
| Normal (18.5–22.9) | 112 (38.1) |
| Overweight (23.0–27.4) | 92 (31.3) |
| Obese (> 27.5) | 67 (22.8) |
| **Comorbidities** | |
| No | 143 (48.6) |
| Chronic lung disease | 2 (0.7) |
| Cardiac disease | 15 (5.1) |
| Morbid obesity | 4 (1.4) |
| Diabetes | 8 (2.7) |
| Liver disease | 4 (1.4) |
| Human immunodeficiency virus infection | 2 (0.7) |
| Other[b] | 32 (10.9) |
| ≥2 comorbidities | 84 (28.6) |
| **Type of cancer** | |
| Cervical | 61 (20.7) |
| Ovary/Peritoneum/Fallopian tube | 115 (39.1) |
| Uterine | 102 (34.7) |
| Vulva | 2 (0.7) |
| Vagina | 6 (2.0) |
| Multiple sites of primary carcinoma | 8 (2.7) |
| **FIGO stage** | |
| I | 137 (46.6) |
| II | 40 (13.6) |
| III | 85 (28.9) |
| IV | 18 (6.1) |
| Recurrent | 14 (4.8) |
| **ECOG Performance status** | |
| 0 | 85 (28.9) |
| 1 | 149 (50.7) |
| 2 | 58 (19.7) |
| 3 | 2 (0.7) |
| **Current status of treatment** | |
| Surveillance/Palliative treatment | 214 (72.8) |
| Ongoing treatment | |
| Chemotherapy | 52 (17.7) |
| Radiation | 12 (4.1) |
| Concurrent chemoradiation | 11 (3.7) |
| Targeted therapy | 2 (0.7) |
| Immunotherapy | 1 (0.3) |
| Hormonal therapy | 2 (0.7) |

*(Continued)*

**Table 1.** (Continued)

Baseline demographic and clinical characteristics of patients with gynecologic cancer who received at least one dose of COVID-19 vaccine. Data are presented as number (percentage) unless otherwise specified. BMI categories are based on Asia-Pacific cutoffs.

a Body mass index based on the cut point of obesity among Asians.

b Of the 32 patients with other comorbidities were osteoporosis, adrenal insufficiency, major depressive disorder, dyslipidemia, hypothyroid and deep vein thrombosis

FIGO: International Federation of Gynecology and Obstetrics; ECOG: Eastern Cooperative Oncology Group Performance Status

**Table 2. Type of COVID-19 vaccination (N = 849).**

| Type of vaccine n (%) | 1st Dose (n = 294) | 2nd Dose (n = 283) | 3rd Dose (n = 190) | 4th Dose (n = 80) | Total n |
|---|---|---|---|---|---|
| Sinovac | 12 (4.1) | 7 (2.5) | 0 (0) | 0 (0) | 19 |
| AstraZeneca | 273 (92.9) | 247 (87.3) | 2 (1) | 0 (0) | 522 |
| Pfizer | 4 (1.4) | 20 (7.0) | 108 (56.3) | 45 (56.3) | 177 |
| Moderna | 5 (1.7) | 9 (3.1) | 82 (42.7) | 35 (43.7) | 131 |

Distribution of COVID-19 vaccine types administered across the first to fourth doses. Data are presented as number (percentage).

than 7 days; no patient developed grade 3–4 symptoms (Fig 1). Detailed adverse events following immunization by vaccine type are presented in S1 Table.

Our study detected COVID-19 infection in 78 patients (27.6%) after injection of the second dose. We observed no association between cancer treatment status and the adverse events of the COVID-19 vaccine and no difference in cancer treatment status after each dose of the vaccine (Table 4).

All the patients who received the first dose of the vaccine were analyzed to determine the factors associated with results that indicated significant vaccine-related adverse events in Table 5. After adjusting for age and BMI, patients younger than 60 years of age were more likely to report vaccine-related adverse events than older patients (adjusted odds ratio 1.99, 95% confidence interval 1.08–3.71; p = 0.029), whereas BMI, comorbidities, type of cancer, stage of cancer, ECOG score, and treatment status were not significantly associated. In the fully adjusted sensitivity analysis (S2 Table), only age < 60 years remained significant (adjusted OR 2.62, 95% CI 1.25–5.47, p = 0.010), and the magnitude and direction of associations were consistent with the primary model, supporting the robustness of the findings.

## Discussion

Our study showed that COVID-19 vaccines are generally well-tolerated in patients with gynecologic cancer. The most common adverse event was pain at the injection site, followed by fever, myalgia, fatigue, and headache. Most vaccine-related adverse events were mild and transient and resolved in a few days without sequelae. The overall incidence of local and systemic adverse events was 80.2%. In comparison, the Pfizer/BioNTech vaccine studies showed incidences of any-grade local and systemic adverse events within 7 days of vaccination of 84.7% and 77.4%, respectively [8]. In the Oxford/AstraZeneca studies, the total incidence of local and systemic adverse events following the first vaccine dose was approximately 61%–88% [9]. Our patient population's response falls within this range, indicating that having a cancer diagnosis and related treatments did not markedly increase the rate of common side effects.

Interestingly, one study patient developed clinical fever with seizures after the first dose of the AstraZeneca vaccine. The patient was subsequently diagnosed with encephalopathy, believe to be precipitated by fever or stress in the context of underlying brain pathology. Several reports have described acute neurologic disorders, such as encephalopathy (hyperacute reversible following unspecified COVID-19 vaccine, post-Moderna), seizures (new-onset refractory status

**Table 3. Both solicited and unsolicited adverse events within 7 days and after 7 days post-vaccination.**

| Side effect | Within 7 days post vaccination n (%) | | | After 7 days post vaccination n (%) | | | P-value[a] |
|---|---|---|---|---|---|---|---|
| | Grade 1 | Grade 2 | Grade 3 | Grade 1 | Grade 2 | Grade 3 | |
| **After the first dose** | | | | | | | <0.001 |
| **Local adverse events** | | | | | | | |
| Injection site reaction | 189 (64.3) | | | 2 (0.6) | | | |
| **Systemic adverse events** | | | | | | | |
| Fever | 130 (44.2) | | | 1 (0.3) | | 1 (0.3) | |
| Headaches | 12 (4.0) | | | | | | |
| Fatigue | 22 (7.5) | 1 (0.3) | | | | | |
| Myalgia | 42 (14.3) | | | | | | |
| Nausea | 1 (0.3) | | | | | | |
| Vomiting | 2 (0.6) | | | | | | |
| Diarrhea | 2 (0.6) | | | | | | |
| Rash | | | | 1 (0.3) | | | |
| Lymphadenopathy | | | | | | | |
| Other | | | | | | 1 (0.3) | |
| **After the second dose** | | | | | | | <0.001 |
| **Local adverse events** | | | | | | | |
| Injection site reaction | 179 (63.3) | | | 1 (0.4) | | | |
| **Systemic adverse events** | | | | | | | |
| Fever | 77 (27.2) | | | | | | |
| Headaches | 7 (2.5) | | | | | | |
| Fatigue | 19 (6.7) | | | | | | |
| Myalgia | 18 (6.4) | 1 (0.4) | | | | | |
| Nausea | 1 (0.4) | | | | | | |
| Vomiting | 1 (0.4) | | | | | | |
| Diarrhea | 2 (0.7) | | | | | | |
| Rash | | | | | | | |
| Lymphadenopathy | | | | | | | |
| **After the third dose** | | | | | | | <0.001 |
| **Local adverse events** | | | | | | | |
| Injection site reaction | 131 (68.9) | | | 2 (1.1) | | | |
| **Systemic adverse events** | | | | | | | |
| Fever | 58 (30.5) | | | | | | |
| Headaches | 4 (2.1) | | | | | | |
| Fatigue | 16 (8.4) | | | | | | |
| Myalgia | 13 (6.8) | | | | | | |
| Nausea | | | | | | | |
| Vomiting | | | | | | | |
| Diarrhea | | | | | | | |
| Rash | 2 (1.1) | | | | | | |
| Lymphadenopathy | | | | 1 (0.5) | | | |
| **After the fourth dose** | | | | | | | <0.001 |
| **Local adverse events** | | | | | | | |
| Injection site reaction | 56 (70) | | | 1 (1.25) | | | |

*(Continued)*

**Table 3.** (Continued)

| Side effect | Within 7 days post vaccination n (%) | | | After 7 days post vaccination n (%) | | | P-value[a] |
|---|---|---|---|---|---|---|---|
| | Grade 1 | Grade 2 | Grade 3 | Grade 1 | Grade 2 | Grade 3 | |
| **Systemic adverse events** | | | | | | | |
| Fever | 21 (26.25) | | | | | | |
| Headaches | | | | | | | |
| Fatigue | 1 (1.25) | | | 1 (1.25) | | | |
| Myalgia | 11 (13.75) | | | | | | |
| Nausea | | | | | | | |
| Vomiting | 8 (10) | | | | | | |
| Diarrhea | | | | | | | |
| Rash | | | | | | | |
| Lymphadenopathy | | | | 1 (1.25) | | | |

Solicited adverse events within 7 days and unsolicited adverse events occurring after 7 days post-vaccination, graded according to CTCAE v5.0. Data are presented as number (percentage). P-values were calculated using Fisher's exact test.

[a]P-values are obtained using the Fisher's exact tests for adverse events within 7 days and after 7 days post-vaccination

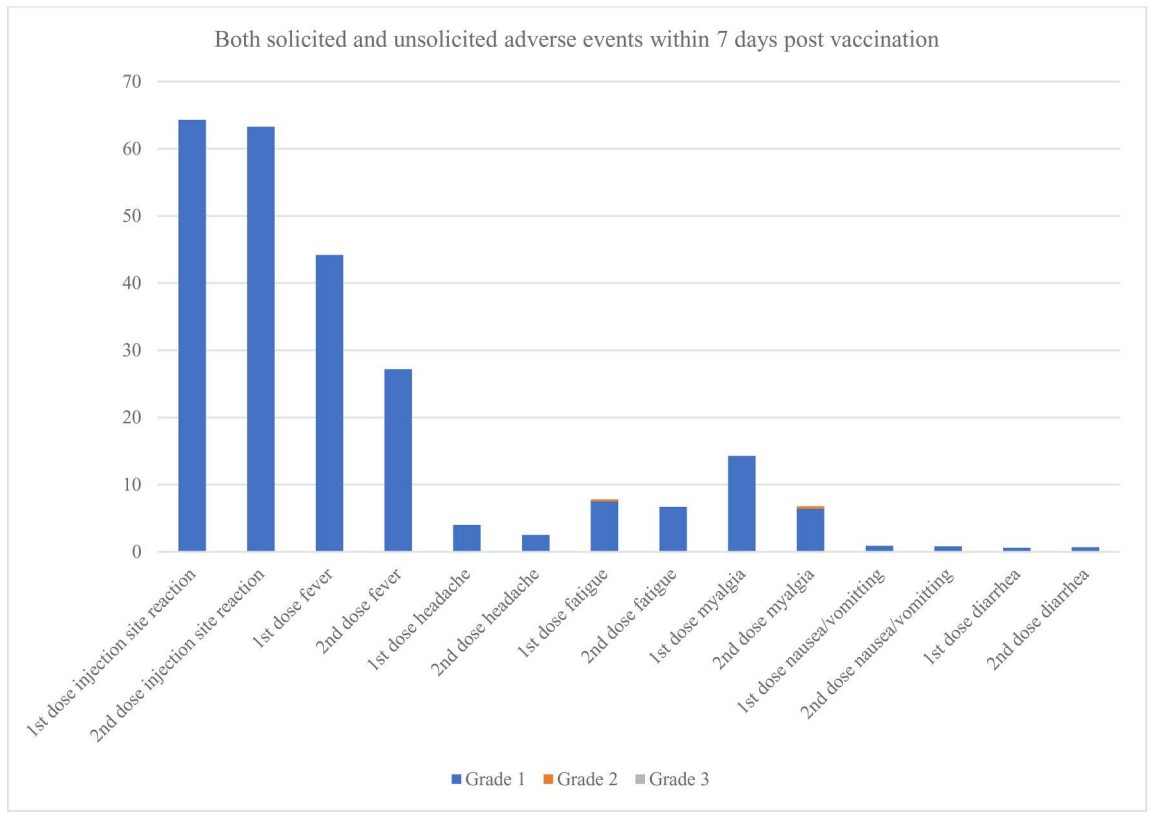

**Fig 1. Both solicited and unsolicited adverse events within 7 days post-vaccination.** Solicited adverse events within 7 days and unsolicited adverse events after 7 days post-vaccination, graded according to CTCAE v5.0.

Table 4. Adverse events of COVID-19 vaccine by current treatment status.

| Treatment | Any grade adverse events | | | | | | | |
|---|---|---|---|---|---|---|---|---|
| | Dose 1 n (%) | P-value[a] | Dose 2 n (%) | P-value[a] | Dose 3 n (%) | P-value[a] | Dose 4 n (%) | P-value[a] |
| **Surveillance/Palliative treatment** **Ongoing treatment** | 169(78.9) | 0.89 | 155 (74.5) | 0.878 | 132 (84.1) | 0.064 | 57 (83.8) | 0.175 |
| Chemotherapy | 44 (84.6) | | 35 (74.5) | | 14 (77.8) | | 5 (100) | |
| Radiation Concurrent | 9 (75.0) | | 8 (66.6) | | 2 (40.0) | | 0 | |
| Concurrent chemoradiation | 9 (81.8) | | 9 (81.8) | | 5 (83.3) | | 4 (100) | |
| Targeted therapy | 2 (100) | | 2 (100) | | 2 (100) | | 1 (100) | |
| Immunotherapy | 1 (100) | | 1 (100) | | 1 (100) | | 1 (100) | |
| Hormonal therapy | 2 (100) | | 2 (100) | | 0 (0) | | 0 (0) | |

Proportion of patients reporting any-grade adverse events following each vaccine dose, stratified by cancer treatment status. Data are presented as number (percentage). P-values were obtained using Fisher's exact test comparing surveillance/palliative versus active treatment groups.

[a] Fisher's exact test was used to compare active treatment versus surveillance/palliative groups. Individual treatment subgroups were not compared separately due to small sample sizes.

epilepticus following the ChAdOx1 nCoV-19, AstraZeneca), acute disseminated encephalomyelitis (following various COVID-19 vaccination), neuroleptic malignant syndrome, and post-vaccine encephalitis, as secondary to the COVID-19 vaccine [10–14]. Although such events are exceedingly uncommon, clinicians should remain vigilant for atypical neurologic symptoms following vaccination, particularly in individuals with pre-existing neurological vulnerabilities. Furthermore, we observed that following the second, third, and fourth doses of the COVID-19 vaccine, the overall incidence of adverse events was not different from that following the first dose. As with the initial vaccination, local adverse events remained the most frequently reported reactions across all subsequent doses. This consistency in the safety profile across multiple vaccine doses is reassuring, particularly for cancer patients who may require ongoing booster vaccinations to maintain adequate immunity. Although most adverse events were mild and self-limiting, their clinical relevance remains important. These findings offer reassurance that vaccination does not interfere with ongoing cancer care, as side effects typically resolved quickly and did not require medical intervention. For patients with cancer—who may express vaccine hesitancy due to concerns about side effects—this data can support informed counseling. Recognizing that younger patients report more symptoms may also help tailor expectations and improve vaccine confidence.

In our study, 27.6% of patients developed COVID-19 after the second vaccine dose. While this figure is similar to a prior report showing a 26% infection rate among vaccinated patients compared with 44.5% among unvaccinated individuals [15], it should be interpreted with caution because national background incidence data for the same period were not available for direct comparison. A multicenter phase 3 trial also demonstrated that vaccine efficacy reached up to 79% in preventing symptomatic COVID-19 at least 14 days after the second dose [16]. The relatively high infection rate observed in our study may be partially attributed to the timing of vaccinations, which occurred during Thailand's mid-2021 surge, dominated by the Delta variant. This variant is known for its higher transmissibility and partial immune escape, particularly in immunocompromised individuals or those with waning immunity. In addition, cancer patients may mount a weaker immune response to vaccination, further increasing their vulnerability. Taken together, these factors provide a plausible explanation for the observed infection rate, although the absence of a direct comparator group limits firm conclusions.

Although chemotherapy-induced myelosuppression extending to both myeloid and lymphoid response can reduce vaccine immunogenicity and reactogenicity [17,18]. However, our study focused primarily on adverse event profiles rather than immunogenicity. We observed no difference in the adverse events of the COVID-19 vaccine between

**Table 5. Logistic regression of the association between demographics, clinical factors and the occurrence of any grade adverse events after 1st dose of COVID-19 vaccine (92.9% AstraZeneca).**

| Characteristic | Univariable analysis OR (95% CI) | P-value | Multivariable analysis Adjusted OR (95% CI) | P-value |
|---|---|---|---|---|
| **Age** | | 0.034 | | 0.029 |
| 18 to <60 years | 1.94 (1.05-3.58) | 0.034 | 1.99 (1.075-3.709) | 0.029 |
| ≥60 years | Reference | 1 | Reference | 1 |
| **Body mass index (kg/m²)[a]** | | 0.164 | | 0.160 |
| Underweight (< 18.5) | 6.99 (0.89-54.30) | 0.063 | 7.37 (0.94-57.58) | 0.057 |
| Normal (18.5–22.9) | Reference | 1 | Reference | 1 |
| Overweight (23.0–27.4) | 1.14 (0.59-2.21) | 0.690 | 1.09 (0.56-2.12) | 0.802 |
| Obese (> 27.5) | 1.81 (0.81-4.03) | 0.146 | 1.76 (0.78-3.93) | 0.171 |
| **Comorbidities** | | 0.641 | | |
| No | Reference | 1 | | |
| Chronic lung disease + Cardiac disease + | 0.89 (0.48-1.62) | 0.696 | | |
| Morbid obesity + Diabetes + Liver disease | | | | |
| + Human immunodeficiency virus infection | | | | |
| Other[b] | 1.46 (0.52-4.10) | 0.472 | | |
| **Type of cancer** | | 0.589 | | |
| Cervical | 0.61(0.28-1.30) | 0.199 | | |
| Ovary/Peritoneum/Fallopian tube | Reference | 1 | | |
| Uterine | 0.81 (0.41-1.62) | 0.555 | | |
| Other gynecologic cancer | 0.60 (0.17-2.04) | 0.408 | | |
| **FIGO stage** | | 0.470 | | |
| I | Reference | 1 | | |
| II | 0.89 (0.37-2.17) | 0.802 | | |
| III | 0.68 (0.35-1.31) | 0.250 | | |
| IV and Recurrent disease | 1.56 (0.50-4.86) | 0.440 | | |
| **ECOG Performance status** | | 0.842 | | |
| 0 | Reference | 1 | | |
| 1 | 0.82 (0.41-1.62) | 0.559 | | |
| 2 + 3 | 0.86 (0.37-1.99) | 0.720 | | |
| **Current status of treatment** | | 0.668 | | |
| Surveillance/Palliative treatment | Reference | 1 | | |
| Ongoing treatment | | | | |
| Chemotherapy | 1.46 (0.64-3.33) | 0.363 | | |
| Radiation | 0.80 (0.21-3.07) | 0.744 | | |
| Others | 1.86 (0.41-8.50) | 0.421 | | |

Univariable and multivariable logistic regression analyses evaluating demographic and clinical factors associated with the occurrence of any-grade adverse events after the first COVID-19 vaccine dose. Results are presented as odds ratios (ORs) with 95% confidence intervals (CIs). BMI categories are based on Asia-Pacific criteria.

[a] Body mass index based on the cut point of obesity among Asians.

[b] Of the 32 patients with other comorbidities were osteoporosis, adrenal insufficiency, major depressive disorder, dyslipidemia, hypothyroid and deep vein thrombosis

patients undergoing active cancer treatment and those under surveillance. These findings are consistent with prior studies reporting no significant difference in side effect profiles across anti-cancer treatment protocols among vaccinated cancer [17–19].

Our analysis showed that older patients (≥ 60 years) had a lower incidence of vaccine-related adverse events compared to younger patients (OR 1.94, 95% confidence interval 1.05–3.58; p = 0.034). This finding may be explained by the higher degree of symptom tolerance in older people and the age-related decline in the immune response. Differences between younger and older healthy adults in the rate of adverse events were previously reported in phase 1–3 clinical trials [4,5,20,21]. In a Pfizer/BioNTech vaccine trial, younger (< 55 years) recipients reported higher rates of local and systemic reactogenicity compared with older (≥ 55 years) recipients after the first vaccine dose [9]. No other independent predictors for developing vaccine-related adverse events were identified.

Furthermore, direct comparisons of AEFI rates between primary (1st/2nd) and booster (3rd/4th) doses should be interpreted cautiously due to marked differences in vaccine platforms used. Approximately 87–93% of 1st and 2nd doses were AstraZeneca (viral vector), while 99% of 3rd/4th doses were mRNA-based (Pfizer/Moderna), reflecting Thailand's national vaccination rollout, which prioritized AZ early, followed by mRNA boosters. These platform differences may influence reactogenicity profiles, precluding general pooled comparisons across phases.

Our study has several limitations. First, its retrospective, observational design and the absence of an unvaccinated control group limit causal interpretation. Second, selection bias may be present, as excluded patients—many of whom were unreachable or vaccinated elsewhere—may have differed in disease stage, treatment status, or vaccine hesitancy, potentially affecting the generalizability of our findings. Third, adverse events were self-reported via telephone interviews, which may have introduced recall bias or underreporting of mild symptoms, despite the use of a structured questionnaire and partial verification through the Mohprompt application. Fourth, although age was adjusted for in our logistic regression model, other potential confounding factors—such as comorbidities, ECOG performance status, and cancer subtypes—were not fully accounted for, limiting the precision of our risk estimates. Fifth, we did not assess immunogenicity, which restricts conclusions regarding vaccine efficacy. Sixth, the single-center setting, predominance of AstraZeneca vaccination, and exclusive focus on gynecologic cancer patients may reduce the broader applicability of our findings. Additionally, most included patients were in remission, had lower ECOG performance status, and earlier FIGO stages, reflecting a healthier cohort, which may have contributed to the lower observed incidence and severity of adverse events. These limitations should be taken into account when interpreting our results. Nonetheless, our findings are consistent with prior studies and support the overall tolerability of COVID-19 vaccination in this patient population.

## Conclusion

Our study findings provide evidence that SARS-CoV-2 vaccination is generally well tolerated in patients with gynecologic cancer, including those actively receiving cancer treatment or those under surveillance. No vaccine-related deaths or life-threatening reactions were observed, and side effect profiles were consistent with those reported in the general population. Younger patients reported more adverse events, likely reflecting a stronger immune response. Although COVID-19 vaccination was generally well tolerated across all doses, these findings should be interpreted with caution because the primary series consisted predominantly of AstraZeneca, while booster doses were almost exclusively mRNA vaccines, which may confound direct comparisons of adverse events following immunization between phases. However, our study did not assess immunogenicity or long-term protection. Therefore, future prospective, multi-center studies are warranted to evaluate vaccine-induced immune responses, durability of protection, and long-term safety outcomes in cancer populations. These findings will be crucial to guiding vaccination strategies and counseling in this vulnerable group.

## Supporting information

**S1 Table. Adverse Events Following Immunization (AEFIs) by Vaccine Type.** Adverse events following immunization (AEFIs) are summarized by COVID-19 vaccine type. Data are presented as number (percentage). Local AEFIs include injection-site reactions, and systemic AEFIs include generalized symptoms. "No side effects" refers to participants reporting no adverse events following vaccination.
(DOCX)

**S2 Table. Fully adjusted logistic regression of factors associated with any grade adverse events after the 1st dose of COVID-19 vaccine.** Multivariable logistic regression analysis was performed to identify factors associated with the occurrence of any-grade adverse events following immunization after the first dose of COVID-19 vaccine. Results are presented as adjusted odds ratios (ORs) with 95% confidence intervals (CIs). P-values <0.05 were considered statistically significant. BMI categories were defined according to Asia-Pacific criteria.
(DOCX)

**S1 File. Interview Questionnaire for Adverse Events after COVID-19 Vaccination.** This file contains the structured interview questionnaire used to collect patient information, COVID-19 vaccination history, and adverse events following immunization. Adverse events were assessed for local and systemic reactions occurring within 7 days after each vaccine dose, as well as delayed adverse events up to 30 days post-vaccination. Severity of symptoms was graded as mild, moderate, or severe based on patient self-report.
(DOCX)

## Acknowledgments

The authors are very grateful to Piyalamporn Havanont (Chulalongkorn University) for her contribution to the statistical analysis. We also thank Anahid Pinchis from Edanz (www.edanz.com/ac) for editing a draft of this manuscript.

## Author contributions

**Conceptualization:** Sasivimon Ratree, Natacha Phoolcharoen.

**Data curation:** Sasivimon Ratree, Nicha Assavapokee.

**Formal analysis:** Sasivimon Ratree, Somsook Santibenchakul.

**Funding acquisition:** Natacha Phoolcharoen.

**Methodology:** Sasivimon Ratree, Somsook Santibenchakul, Natacha Phoolcharoen.

**Supervision:** Nakarin Sirisabya, Waranyoo Phoolcharoen.

**Writing – original draft:** Sasivimon Ratree, Natacha Phoolcharoen.

**Writing – review & editing:** Nicha Assavapokee, Nakarin Sirisabya, Waranyoo Phoolcharoen, Natacha Phoolcharoen.

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
