## [Decision Letter · Decision Letter 0]

29 Mar 2025

PONE-D-24-54040Safety of COVID-19 Vaccination in gynecologic cancer patients  in ThailandPLOS ONE

Dear Dr. Phoolcharoen,

Thank you for submitting your manuscript to PLOS ONE. After careful consideration, we feel that it has merit but does not fully meet PLOS ONE’s publication criteria as it currently stands. Therefore, we invite you to submit a revised version of the manuscript that addresses the points raised during the review process.

We look forward to receiving your revised manuscript.

Kind regards,

Mohammad Barary

Academic Editor

PLOS ONE

Journal Requirements:

Fundamental fund no. HEAF67300053

5. Please note that your Data Availability Statement is currently missing [the repository name and/or the DOI/accession number of each dataset OR a direct link to access each database]. If your manuscript is accepted for publication, you will be asked to provide these details on a very short timeline. We therefore suggest that you provide this information now, though we will not hold up the peer review process if you are unable.

6. Please amend either the abstract on the online submission form (via Edit Submission) or the abstract in the manuscript so that they are identical.

7. Please remove all personal information, ensure that the data shared are in accordance with participant consent, and re-upload a fully anonymized data set.

Reviewers' comments:

Reviewer's Responses to Questions

**Comments to the Author**

1. Is the manuscript technically sound, and do the data support the conclusions?

Reviewer #1: No

Reviewer #2: Yes

2. Has the statistical analysis been performed appropriately and rigorously?

Reviewer #1: No

Reviewer #2: Yes

3. Have the authors made all data underlying the findings in their manuscript fully available?

Reviewer #1: Yes

Reviewer #2: Yes

4. Is the manuscript presented in an intelligible fashion and written in standard English?

Reviewer #1: Yes

Reviewer #2: Yes

5. Review Comments to the Author

Reviewer #1: Summary: This descriptive study assesses the prevalence of adverse events following immunization (AEFI) after receipt of COVID-19 vaccines in patients with gynecologic cancers in Thailand. Specifically, the authors assess AEFI within 7 days of vaccination, between 7 and 30 days, and post-vaccination COVID-19 infection 14 days after receiving a second dose. This study aims to add to the literature on the safety of the vaccine in immunocompromised patients. Increasing confidence in COVID-19 vaccines is a crucial need, particularly for immunocompromised patients and the insights from this study could help inform this need. However, the reviewer has concerns about the authors interpretation of the inferential statistics included in the manuscript. The reviewer suggests restructuring in some areas, clarification of methods and interpretation, and caution with interpretation. Below are questions/comments about the contents of the manuscript.

Major

- What is the rationale for specifically looking at AEFIs in gynecologic cancer patients? Do they simply represent a subset of immunocompromised patients?

- The manuscript makes definitive claims that are not possible by the analysis. For example, “This study determines the safety of the COVID-19 vaccine in vaccinated patients with 10 gynecologic cancer by assessing side effects and side effect severity.” However, this is not vaccine safety study but rather a descriptive study of reported AEFIs in vaccinated patients. The authors should be careful with language that can be interpreted causally. It would be more accurate to say that “this study evaluates the prevalence of reported adverse events following receipt of COVID-19 vaccine.”

- Of the 846 patients with gynecologic cancers, 294 were eligible for the analysis. Thus 65% of the initial patient population is excluded from the analysis.

o First, the authors should provide a flowchart to specify the proportions of the original study population excluded from the analytic sample and the various reasons (e.g. n = xx were excluded for missing data, n = xx were excluded because age was less than 18 years of age etc...)

o Second, do the authors have information on the excluded participants? Are they significantly different from the included participants? Are there greater proportions of excluded patients with ongoing treatment, greater FIGO staging than the included patients. Differences such as these present an opportunity for selection bias in these analyses. In this study, it is plausible that both the exposure and outcome are associated with selection into the study. This would result in selection bias.

Please note that if the authors are strictly interested in a descriptive study, the selection bias does not need to be addressed. However, since the authors seem to be interested in inferential statistics and have commented on vaccine safety, then they should address the potential selection bias, the magnitude and direction based on the selection proportions in this study

- The authors should expand the limitations section:

o 1) The AEFI were self-reported through telephone interviews and may be prone to misclassification. For example, were patients asked about temperature if they had a fever or were they asked to report whether or not they had a fever? The latter would result in lower specificity and increase false negatives

o 2) The authors allude to recall bias because some patients have contemporaneous entries of symptoms and others do not. The proposed mechanism of recall bias would be that patients who entered information in the app have better recall than patients who do not. The authors should clarify this if that is in fact what they are alluding to in the manuscript.

o 3) Please see previous comments about selection bias

- Since the majority of patients in the study are in remission, with lower ECOG performance status scores and FIGO stage, can the authors comment on the possibility that the patients included in their study are healthier and the potential impact on their results? Particular since the authors observe lower odds of adverse events in people with co-morbidities…

- For the analyses examining any grade AEFI by treatment status in Table 4, can the authors specify the comparisons for the Fisher’s exact tests? The table is difficult to understand and interpret. Did the authors dichotomize treatment status into ongoing vs. palliative when running Fisher’s exact test?

o If the authors used the individual treatment categories, how did they deal with categories with fewer than 2 observations (MLE for the Fisher’s test in that analysis would probability of adverse events predicted on treatment status/1-probability of adverse events predicted on treatment status which would result in dividing by 0 in some categories, and there be undefined)

o The authors mention using Chi-square and Fisher’s exact test in the methods section but it looks like they only report results from Fisher’s exact tests. Did the authors use both for the same analyses or used different methods for different analyses? If the former, why? Was it because chi-square relies on approximation while Fisher’s provides exact p-values (and more conservative)?

- Do the authors have information on timing between the doses? It would be informative to know the mean or median time between 1st/2nd/3rd/4th doses in this study population.

- Based on table 2, it seems that there is almost a bimodal distribution of exposures where 1st and 2nd dose are ~90% Sinovac and AstraZeneca while 3rd and 4th doses are ~90% Moderna and Pfizer. Since these vaccines have different mechanisms to illicit immune reactions, did the authors consider separating their analyses to look at AEFIs following specific vaccines rather than collapsing by dose received?

Minor

- Line 18 – There is a typo in this sentence….” between January 2020 and August 2021 were reviewed was a descriptive…”

Reviewer #2: Reviewer comments and suggestions regarding the manuscript entitled “Safety of COVID-19 Vaccination in gynecologic cancer patients in Thailand” with the manuscript number “PONE-D-24-54040.”

This manuscript reports on a descriptive, single-center study conducted at King Chulalongkorn Memorial Hospital in Thailand. The study assessed the safety and tolerability of COVID-19 vaccines in 294 gynecologic cancer patients by collecting adverse event data through telephone interviews. The most common adverse events were mild to moderate injection site reactions, fever, myalgia, fatigue, and headache. Logistic regression analysis revealed that patients under 60 years of age had nearly twice the odds of experiencing adverse events compared to older patients, while cancer treatment status did not significantly affect the rate of adverse events. Additionally, the study provides data on COVID-19 infections after two doses of vaccination, with an observed infection rate of 27.6%.

The paper can make an essential contribution to its field, but authors should address some significant considerations:

I. Title:

The precise title reflects the study's focus on vaccine safety in gynecologic cancer patients in Thailand.

II. Abstract:

The abstract concisely presents the study’s objectives, methods, key results, and conclusions. It effectively highlights that adverse events are mostly mild and that younger patients report more side effects. Minor improvements could include briefly mentioning the statistical methods used to reinforce the study's rigor.

III. Introduction:

The introduction provides a transparent background on COVID-19, the increased risks for cancer patients, and the importance of assessing vaccine safety in this group. It adequately sets the context by discussing the types of vaccines used in Thailand and the existing literature gaps. A stronger emphasis on how this study advances current knowledge would further highlight its novelty.

IV. Materials and Methods:

• The study’s single-center, descriptive design may limit the generalizability of the findings. The authors should discuss how representative their patient population is compared to the broader population of gynecologic cancer patients in Thailand.

• The study period is reported as January 2020 to August 2021. Given that COVID-19 vaccines became widely available only after late 2020 in many regions, clarification is needed on the timeline of vaccine administration in this setting.

• Data on adverse events were collected via telephone interviews, introducing potential recall bias and subjectivity. Details on the interview process—including whether a standardized questionnaire was used and the training of interviewers—should be provided.

• Information on the response rate and how missing or incomplete data were handled would strengthen the validity of the study findings.

• While adverse events were graded using the Common Terminology Criteria for Adverse Events (CTCAE v5.0), the manuscript would benefit from more explicit descriptions regarding whether events were solicited or unsolicited and the exact time points at which these events were recorded.

• The study involves multiple vaccine types (Oxford/AstraZeneca, Sinovac, Moderna, and Pfizer/BioNTech). A stratified analysis by vaccine type, including potential differences in adverse event profiles, could provide more nuanced insights.

• Logistic regression identified age as a significant predictor of adverse events. However, the analysis should address potential confounding factors such as comorbidities, ECOG performance status, and specific cancer subtypes.

• Given that a substantial proportion of patients were either in remission/surveillance or undergoing active treatment, further subgroup analyses would help clarify whether treatment status truly has no impact on vaccine safety.

V. Results:

• The reported overall incidence of adverse events (80.2%) is within the range observed in other studies; however, the clinical relevance of predominantly mild events should be discussed in more depth.

• An infection rate of 27.6% after two vaccine doses is relatively high. The authors should discuss factors such as the timing of infection relative to vaccination and circulating variants during the study period, and how this compares with infection rates in non-cancer populations.

• The finding that patients under 60 report more adverse events is consistent with prior vaccine trials, but additional discussion on the biological or immunological rationale behind this observation would be valuable.

VI. Discussion:

• The discussion addresses some limitations, including the reliance on telephone interviews and potential recall bias. It would be helpful also to consider selection bias (e.g., which patients were reachable by phone) and the absence of objective immunogenicity assessments.

• While the manuscript concludes that COVID-19 vaccines are safe in this patient population, further discussion of the clinical implications, particularly regarding vaccine hesitancy and patient counseling, is warranted.

VII. Conclusion:

To validate these findings, the authors should suggest prospective studies or multi-center collaborations that include immunogenicity and longer-term outcome measures.

VIII. Tables and Figures:

The tables are well-structured and delineate key findings, such as demographic distributions, vaccination types, and adverse event rates. Figures summarizing adverse event trends complement the data presentation and facilitate a visual understanding of the temporal patterns. The statistical analyses are appropriately reported, highlighting significant findings (e.g., higher adverse events in younger patients).

6. PLOS authors have the option to publish the peer review history of their article (what does this mean?). If published, this will include your full peer review and any attached files.

Reviewer #1: No

Reviewer #2: No

---

## [Author Response · Author response to Decision Letter 1]

19 Jun 2025

We have carefully addressed all reviewer and editor comments in the attached point-by-point response document, with corresponding revisions clearly marked in the revised manuscript. We appreciate the constructive feedback, which has helped improve the clarity and quality of our work.

---

## [Editor Report · Decision Letter 1]

6 Aug 2025

PONE-D-24-54040R1Reported adverse events following COVID-19 vaccination in gynecologic cancer patients in Thailand: a descriptive studyPLOS ONE

Dear Dr. Phoolcharoen,

Thank you for submitting your manuscript to PLOS ONE. After careful consideration, we feel that it has merit but does not fully meet PLOS ONE’s publication criteria as it currently stands. Therefore, we invite you to submit a revised version of the manuscript that addresses the points raised during the review process.

We look forward to receiving your revised manuscript.

Kind regards,

Mohammad Barary, MD

Academic Editor

PLOS ONE

Journal Requirements:

Additional Editor Comments:

Thank you for the revised submission of your manuscript. It has improved significantly, but some methodological and analytical concerns must be resolved before the study meets PLOS ONE standards:

1- Confounder control in logistic regression

Table 5 adjusts only for age and BMI; other clinically plausible covariates (comorbidities, ECOG, cancer stage, treatment status) were excluded despite their prevalence and potential to confound adverse-event reporting. Either present a formal rationale (e.g., events-per-variable calculation) for the reduced model or add a fully adjusted sensitivity analysis that includes the major covariates.

2- Definition and ascertainment of adverse events

Adverse events were collected by telephone without objective confirmation; fever and other systemic symptoms relied on patient recall, supplemented by “Mohprompt” app entries. Provide the full interview questionnaire as Supplementary Material. Specify how grades (especially fever) were assigned when no temperature was recorded.

3- Interpretation of post-dose-2 infection rate

Seventy-eight of 283 patients (27.6%) developed PCR-confirmed COVID-19 ≥14 days after dose 2. This figure is interpreted without reference to background incidence. Add Thai incidence data for the corresponding period, or temper the interpretation and clearly acknowledge the absence of a comparator.

---

## [Author Response · Author response to Decision Letter 2]

16 Aug 2025

Response to Reviewer

Comment 1: 1- Confounder control in logistic regression

Table 5 adjusts only for age and BMI; other clinically plausible covariates (comorbidities, ECOG, cancer stage, treatment status) were excluded despite their prevalence and potential to confound adverse-event reporting. Either present a formal rationale (e.g., events-per-variable calculation) for the reduced model or add a fully adjusted sensitivity analysis that includes the major covariates.

Response: We appreciate the reviewer’s insightful comment. In our primary logistic regression model, we included age and BMI as covariates because of their established biological plausibility and to preserve the stability of estimates given the limited number of adverse events relative to sample size. To address the reviewer’s concern, we have now conducted a fully adjusted sensitivity analysis that included additional clinically relevant covariates (comorbidities, cancer type, FIGO stage, ECOG performance status, and treatment status). The results of this sensitivity analysis are presented in Supplementary Table S2. The findings confirmed that only age <60 years remained significantly associated with the occurrence of any-grade adverse events after the first vaccine dose (adjusted OR 2.62, 95% CI 1.25–5.47, p = 0.010). Other covariates were not statistically significant, and the direction and magnitude of associations were consistent with the primary model, supporting the robustness of our main findings (Page17, Lines230-233).

2- Definition and ascertainment of adverse events

Adverse events were collected by telephone without objective confirmation; fever and other systemic symptoms relied on patient recall, supplemented by “Mohprompt” app entries. Provide the full interview questionnaire as Supplementary Material. Specify how grades (especially fever) were assigned when no temperature was recorded.

Response: We have now included the full structured interview questionnaire used to collect adverse event data as Supplementary Material (Supplementary File 3). Adverse events were graded according to the Common Terminology Criteria for Adverse Events (CTCAE) version 5.0. For fever, if patients did not recall the exact temperature, grading was based on the patient’s description of severity: (1) “low-grade or mild fever” was classified as Grade 1, (2) “moderate fever” interfering with daily activity as Grade 2, and (3) “severe fever requiring medical attention or hospitalization” as Grade 3. Other systemic symptoms (fatigue, myalgia, headache, etc.) were also graded using CTCAE v5.0 definitions based on interference with daily function, even in the absence of objective measurement. We have revised the Methods section accordingly (Page 7, Lines 121-128).

3- Interpretation of post-dose-2 infection rate

Seventy-eight of 283 patients (27.6%) developed PCR-confirmed COVID-19 ≥14 days after dose 2. This figure is interpreted without reference to background incidence. Add Thai incidence data for the corresponding period, or temper the interpretation and clearly acknowledge the absence of a comparator.

Response: We thank the reviewer for this important suggestion. Unfortunately, background national COVID-19 incidence data specific to our study population and time frame are not readily available for direct comparison. We have therefore tempered our interpretation and explicitly acknowledged the absence of a comparator in the revised Discussion section (Page 10, Lines 281-282 and Page 21, Line 292).

---

## [Decision Letter · Decision Letter 2]

26 Dec 2025

PONE-D-24-54040R2Reported adverse events following COVID-19 vaccination in gynecologic cancer patients in Thailand: a descriptive studyPLOS One

Dear Dr. Phoolcharoen,

Thank you for submitting your manuscript to PLOS ONE. After careful consideration, we feel that it has merit but does not fully meet PLOS ONE’s publication criteria as it currently stands. Therefore, we invite you to submit a revised version of the manuscript that addresses the points raised during the review process.

We look forward to receiving your revised manuscript.

Kind regards,

Omnia S. El Seifi, M.D., Ph.D.

Academic Editor

PLOS One

Journal Requirements:

Reviewers' comments:

Reviewer's Responses to Questions

**Comments to the Author**

1. If the authors have adequately addressed your comments raised in a previous round of review and you feel that this manuscript is now acceptable for publication, you may indicate that here to bypass the “Comments to the Author” section, enter your conflict of interest statement in the “Confidential to Editor” section, and submit your "Accept" recommendation.

Reviewer #3: (No Response)

2. Is the manuscript technically sound, and do the data support the conclusions?

Reviewer #3: Partly

3. Has the statistical analysis been performed appropriately and rigorously?

Reviewer #3: Yes

4. Have the authors made all data underlying the findings in their manuscript fully available?

Reviewer #3: No

5. Is the manuscript presented in an intelligible fashion and written in standard English?

Reviewer #3: Yes

6. Review Comments to the Author

Reviewer #3: This is a useful paper addresing the AEFI following Covid-19 vaccine in a special group. I have some comments to author as follows:

1) Due to a significant different proportion of Covid-19 vaccine uses during the phase of 1st&2nd doses ( around 90% AZ), and 3rd&4th doses (around 99% mRNA), so AEFI from these two phases should not be compared together in general manner; different proportion of vaccine use should be addressed in the discussion , and also in the conclusion.

2) Please give the detail for one patient who developed fever and seizure after 1st dose of AZ; was Vaccine-Induced Thrombotic Thrombocytopenia (VITT) was excluded? and what about the final clinical outcome of this patient.

3) Table 5 - its title should indicate : Logistic regression...........after 1st dose of COVID-19 vaccine ( 93% AZ).

4) Please give detail for abbreviations: FIGO, ECOG in the manuscript.

5) Line 261 - should mention about the type of Covid 19 vaccines used in those studies; Ref 10-14

7. PLOS authors have the option to publish the peer review history of their article (what does this mean?). If published, this will include your full peer review and any attached files.

Reviewer #3: No

---

## [Author Response · Author response to Decision Letter 3]

28 Dec 2025

Comment 1

Due to a significant different proportion of Covid-19 vaccine uses during the phase of 1st&2nd doses (around 90% AZ), and 3rd&4th doses (around 99% mRNA), so AEFI from these two phases should not be compared together in general manner; different proportion of vaccine use should be addressed in the discussion, and also in the conclusion.

Response:

This comment has been fully addressed in both the Discussion and Conclusion.

Discussion section – new paragraph added

“Furthermore, direct comparisons of AEFI rates between primary (1st/2nd) and booster (3rd/4th) doses should be interpreted cautiously due to marked differences in vaccine platforms used. Approximately 87-93% of 1st and 2nd doses were AstraZeneca (viral vector), while 99% of 3rd/4th doses were mRNA-based (Pfizer/Moderna), reflecting Thailand's national vaccination rollout, which prioritized AZ early, followed by mRNA boosters. These platform differences may influence reactogenicity profiles, precluding general pooled comparisons across phases.” Line 323-330, Page 23.

Conclusion section – clarifying sentence added

“Although COVID‑19 vaccination was generally well tolerated across all doses, these findings should be interpreted with caution because the primary series consisted predominantly of AstraZeneca, while booster doses were almost exclusively mRNA vaccines, which may confound direct comparisons of adverse events following immunization between phases.” Line 359-364, Page 24-25.

Comment 2

Please give the detail for one patient who developed fever and seizure after 1st dose of AZ; was Vaccine-Induced Thrombotic Thrombocytopenia (VITT) was excluded? and what about the final clinical outcome of this patient.

Response:

We have expanded the Results to provide a clear description of this case, including explicit exclusion of VITT and the clinical outcome.

“This patient was diagnosed with encephalopathy attributed to stress or fever in the context of underlying pathologic brain changes from an old ischemic stroke. Vaccine-Induced Immune Thrombotic Thrombocytopenia (VITT) was systematically excluded through normal platelet count (>150 × 10⁹/L), absence of thrombosis on imaging (brain CT and venous Doppler), normal D-dimer, and negative anti-platelet factor 4, antibody enzyme-linked immunosorbent assay (ELISA), confirming non-thrombotic etiology per international diagnostic criteria. The patient fully recovered after supportive care (antipyretics, anticonvulsants, and neurology consultation) with no neurologic sequelae at 6-month follow-up.” Line 195-203, Page 12-13.

Comment 3

Table 5 - its title should indicate : Logistic regression...........after 1st dose of COVID-19 vaccine ( 93% AZ).

Response:

We have therefore revised the title of Table 5 to reflect the composition of the first‑dose cohort explicitly.

Revised Table 5 title:

“Table 5. Logistic regression of the association between demographics, clinical factors, and the occurrence of any‑grade adverse events after the first dose of COVID‑19 vaccine (92.9% AstraZeneca)” on Page 18.

Comment 4

Please give detail for abbreviations: FIGO, ECOG in the manuscript.

Response:

We have added full terms to the footnote of Table 1 on page 10:

“FIGO, International Federation of Gynecology and Obstetrics; ECOG, Eastern Cooperative Oncology Group performance status

Comment 5

Line 261 - should mention about the type of Covid 19 vaccines used in those studies; Ref 10-14.

Response:

The Discussion section referencing neurologic adverse events from the literature has been revised to specify the vaccine types associated with each cited report, based on the original articles.

The relevant sentence now reads:

“Several reports have described acute neurologic disorders, such as encephalopathy (hyperacute reversible following unspecified COVID-19 vaccine , post-Moderna), seizures (new-onset refractory status epilepticus following the ChAdOx1 nCoV‑19, AstraZeneca), acute disseminated encephalomyelitis (following various COVID-19 vaccination), neuroleptic malignant syndrome, and post-vaccine encephalitis, as secondary to the COVID-19 vaccine”. Line 268-274, page 20.

---

## [Decision Letter · Decision Letter 3]

20 Jan 2026

Reported adverse events following COVID-19 vaccination in gynecologic cancer patients in Thailand: a descriptive study

PONE-D-24-54040R3

Dear Dr. Phoolcharoen,

We’re pleased to inform you that your manuscript has been judged scientifically suitable for publication and will be formally accepted for publication once it meets all outstanding technical requirements.

Kind regards,

Omnia S. El Seifi, M.D., Ph.D.

Academic Editor

PLOS One

Additional Editor Comments (optional):

Reviewers' comments:

Reviewer's Responses to Questions

**Comments to the Author**

1. If the authors have adequately addressed your comments raised in a previous round of review and you feel that this manuscript is now acceptable for publication, you may indicate that here to bypass the “Comments to the Author” section, enter your conflict of interest statement in the “Confidential to Editor” section, and submit your "Accept" recommendation.

Reviewer #3: All comments have been addressed

2. Is the manuscript technically sound, and do the data support the conclusions?

Reviewer #3: Yes

3. Has the statistical analysis been performed appropriately and rigorously?

Reviewer #3: Yes

4. Have the authors made all data underlying the findings in their manuscript fully available?

Reviewer #3: Yes

5. Is the manuscript presented in an intelligible fashion and written in standard English?

Reviewer #3: Yes

6. Review Comments to the Author

Reviewer #3: For the title - It would be good to add "year of study" .

For the abstract - if no word count limited, it would be good to address ; 1) mRNA vaccine as the majority for booster dose, 2) majority of the cohort represent the patient in remission with stable condition.

7. PLOS authors have the option to publish the peer review history of their article (what does this mean?). If published, this will include your full peer review and any attached files.

Reviewer #3: No

---

## [Editor Report · Acceptance letter]

PONE-D-24-54040R3

PLOS One

Dear Dr. Phoolcharoen,

I'm pleased to inform you that your manuscript has been deemed suitable for publication in PLOS One. Congratulations! Your manuscript is now being handed over to our production team.

Kind regards,

on behalf of

Professor Omnia S. El Seifi

Academic Editor

PLOS One